# Immunological Memory of Psoriatic Lesions

**DOI:** 10.3390/ijms21020625

**Published:** 2020-01-17

**Authors:** Agnieszka Owczarczyk-Saczonek, Magdalena Krajewska-Włodarczyk, Marta Kasprowicz-Furmańczyk, Waldemar Placek

**Affiliations:** 1Department of Dermatology, Sexually Transmitted Diseases and Clinical Immunology, The University of Warmia and Mazury, Al. Wojska Polskiego 30, 10-229 Olsztyn, Poland; martak03@wp.pl (M.K.-F.); w.placek@wp.pl (W.P.); 2Department of Rheumatology, Municipal Hospital in Olsztyn, 10-229 Olsztyn, Poland; magdalenakw@op.pl; 3Department of Internal Medicine, School of Medicine, Collegium Medicum, University of Warmia and Mazury, 10-900 Olsztyn, Poland

**Keywords:** psoriasis, tissue resident memory cells, IL-17

## Abstract

The natural course of psoriasis is the appearance of new lesions in the place of previous ones, which disappeared after a successful therapy. Recent studies of psoriasis etiopathogenesis showed that after psoriatic plaques have disappeared, in healthy skin we can still find a trace of inflammation in the form of tissue resident memory cells (TRM). They are originally responsible for protection against viral and bacterial infections in non-lymphatic tissues. In psoriatic inflammation, they are characterized by heterogeneity depending on their origin. CD8+ T cells TRM are abundantly present in psoriatic epidermis, while CD4+ TRM preferentially populate the dermis. In psoriasis, epidermal CD8+ TRM cells express CLA, CCR6, CD103 and IL-23R antigen and produce IL-17A during ex vivo stimulation. However, CD4+ CD103+ TRM can also colonize the epidermis and produce IL-22 during stimulation. Besides T cells, Th22 and epidermal DCs proved that epidermal cells in healed skin were still present and functioning after several years of disease remission. It explains the clinical phenomenon of the tendency of psoriatic lesions to relapse in the same location and it allows to develop new therapeutic strategies in the future.

## 1. Introduction

The natural course of psoriasis is the long-term persistence of lesions in the same anatomical regions and the appearance of new ones in places where they have resolved after successful therapy. The appearance of new lesions in the place of previous ones, which disappeared after a successful therapy, evokes feelings of depression, hopelessness and depression in patients and induces conviction in the lack of effective therapies. Recent studies of psoriasis etiopathogenesis showed that after psoriatic plaques have disappeared, in healthy skin we can still find a trace of inflammation in the form of tissue resident memory cells (TRM). They are able to initiate an inflammatory cascade and cause a relapse in the same location. In psoriasis, even during remission, they become a source of important inflammatory cytokines IL-17A and IL-22 [1,2,3,4]. T cell-associated gene (*LCK* and *TRCB1*) and inflammatory gene (*IL17*, *IL22* and *IFN-γ*) activity remains increased in the skin after psoriasis lesions, at least three months after the onset of the treatment with TNF-α inhibitors [5,6]. However, not only TRMs are responsible for clinical relapses at the same location (Figure 1) (Table 1). This article is a review of current knowledge about the “immune memory” of psoriatic lesions.

## 2. Tissue Resident Memory Cells (TRM)

TRM cells are transcriptionally, phenotypically and functionally different from traditional central memory cells (TCM) and effector memory T cells (TEM). They recirculate among blood, T cell zones of secondary lymphoid organs, lymph and non-lymphoid tissues, such as the skin, where they act as alarm sensors or cytotoxic killers. They are characterized by heterogeneity depending on their origin [5,7,8,9,10,11].

They are originally responsible for protection against viral and bacterial infections in non-lymphatic tissues. The protective activity of TRM has been confirmed in HSV infection, vaccinia pox, lymphocytic choriomeningitis virus (LCMV), influenza, listeriosis, malaria, as well as in some types of cancer. The vaccination task is, among others, the education of TRM to react very quickly against an infection [9]. TRM cells are locally found in many tissues providing a rapid in situ response to infectious agents more effectively than T cells of effector memory. They also secrete granzyme B, which helps to reduce the spread of pathogens at the site of infection. They can activate the innate and adaptive mechanisms of the immune response [2,10,12,13,14,15]. TRM acts as a bridge between the adaptive and innate immune systems. Because many viruses have tissue tropism, TRM also provides protective immune responses for tissue that it has previously encountered. Since, HSV-specific TRMs have been found in the skin, rotavirus-specific TRMs in the intestines and flu-specific TRMs in the lungs [16,17]. However, in addition to their protective functions, there is more and more evidence of their involvement in the pathogenesis of autoimmune disorders such as psoriasis, vitiligo, autoimmune hepatitis, rheumatoid arthritis and lymphomas [11].

The main surface markers of human TRM cells are CD49a, CD69 and CD103. The CD103 marker (integrin αE subunit) is only expressed on CD8+ TRM cells, not CD4+ TRM. Its expression is most prominent on epidermal CD4+ and CD8+ TRM cells because it enables TRM binding to E-cadherin, which is widely expressed by epithelial cells [8,13,18]. The CD69 marker plays a key role in distinguishing T cells in tissues from those in circulation and it is responsible for the colonization of these cells in tissues, preventing them from recirculating. CD69 binds and inhibits the expression of the cell surface of sphingosine-1-phosphate receptor (S1PR1). The transcription factor KLF2 promotes the expression of S1PR1, thereby CD69 and may allow recirculation of TRM in unusual situations [19]. S1PR1 reacts to the S1P ligand, which is released by endothelial cells in the blood and lymph, allowing TRM cells to leave the tissues for circulation. This situation is observed, among others, after exposure to an allergen resulting in formation of secondary TRM cells [19]. However, the latest findings indicate that the importance of CD69 for TRM residence may depend on the tissue in which they are located (essential for TRM in the kidneys, no relevance in the gut) [18]. However, expression levels may vary between T cells in different tissues. Another marker that can be used to differentiate two subsets of TRM cells is CD49a—the α-subunit of the α1β1 integrin receptor (also known as very late antigen VLA-1). It determines a subset of CD8+ TRM cells that are localized in the epidermis. These cells produce IFN-γ and acquire high cytotoxic capacity upon IL-15 stimulation [20]. CD8+ CD49a+ TRM cells produce perforin and IFN-γ, which is a key cytokine in fighting viral infections. CD8+ CD49-TRM cells produce IL-17 [20,21]. TRM also expresses the Program Cell Death Protein-1 (PD-1) and T-cell immunoglobulin mucin receptor 3 (TIM3), which are the surface proteins that suppress T cell activity. Their expression occurs in inflamed tissues. In such situations, TRMs have an anti-inflammatory effect, which indicates the ambiguous nature of these cells [22,23].

In healthy individuals, CD8^+^CD103^+^CD49a^−^TRM cells were present in both dermis and epidermis, whereas CD8^+^CD103^+^CD49a^+^TRM cells were localized in the epidermis. Epidermal CD8^+^CD103^+^CD49a^−^TRM cells were predominant IL-17 producers, whereas CD8^+^CD103^+^CD49a^+^TRM cells excelled at IFN-γ production and rapidly gained a cytotoxic capacity following IL-15 stimulation [21].

TRM cells develop from circulating effector T cell precursors in response to an antigen. CD103 plays a major role in their formation, and the expression of this integrin depends on the TGF-β cytokine. CD8+ T effector cells, which do not produce TGF-β, have no expression of CD103 and they do not differentiate into TRM cells. The homeostatic IL-15 cytokine, pro-inflammatory cytokines (IL-12 and IL-18) and barrier cytokines, such as IL-33, which support TRM cell formation and survival, play an important role in the development of TRM cells [7,8].

The second type of TRM are CD4+ cells, which are less-known ones, and are located near the vessels in the dermis; they do not express CD103 and have a high proliferative capacity. They are responsible for defense against *Leishmania* and *Candida albicans* in the gut, lungs and reproductive system [24].

## 3. Tissue Resident Memory Cells in Psoriasis

CD4+ T cells with the Th1 and Th17 phenotype have long been considered the main pathogenic subpopulations of T cells. LL37 (antimicrobial peptide derived from keratinocytes) and ADAMTSL5 (protein produced by melanocytes) in the pathogenesis of the disease has been more appreciated. The abovementioned proteins are considered as autoantigens in psoriasis. Similarly, CD4 T-cells recognize LL37 as an autoantigen and they correlate with PASI [25,26]. CD8+ T cells with resident memory tissue phenotype (TRM) are abundantly present in psoriatic epidermis, while CD4+ TRM preferentially populate the dermis. The differences in colonization result from the expression of CD69, which blocks the sphingosine-1-phosphate receptor (S1P1), a receptor normally allowing lymph entry. In addition, a significant proportion of skin TRM expresses CD103, the αEβ7 integrin chain, which interacts with E-cadherin expressed by keratinocytes (Figure 2). The signal required for their residence is TGF-β via TGF-βRII [9]. Hair follicles through IL-15 and IL-7 production are also important in their recruitment [27]. In addition to cytokines, lipids available in the skin are essential to maintain TRM [10,17]

TRMs have been described in unchanged, healed skin in places of recurrent psoriasis, which indicates their role in the disease’s local memory [28]. Recirculation of memory T cells between the skin and circulation is a newly recognized immunological mechanism that plays an important role in the initiation of psoriasiform response [3,29,30] (Figure 3). Their particular pathogenicity is determined by the fact that they have the ability to produce IL-17 and IL-22, pro-inflammatory, key cytokines involved in this process [23,29].

The first study that highlighted the role of resident skin T cells in psoriasis was the experiment of Boyman et al. [31]. They implanted unchanged human psoriatic skin to immunodeficient mice (AGR129 mice, deficient in type I and II interferon receptors and the recombination activating gene 2). The reproduction of resident human T cells and the formation of psoriatic lesions were observed within eight weeks. They showed the CD8+ phenotype with TNF-α susceptibility and were located mainly in the epidermis and the dermal–epidermal junction [31].

Diani et al. [1] analyzed the phenotype of circulating T cells in patients with psoriasis with the assessment of gene expression in psoriatic skin. It was found that circulating CCR6+ CD4+ TEM (effector memory cells) and T effector cells significantly correlated with the severity of skin lesions and inflammation (CRP) while the percentage of CXCR3+ CD4+ TEM cells correlated negatively. In addition, CLA+ CD4+ TCM cells expressing CCR6+ or CCR4+ CXCR3+ negatively correlated with the severity of psoriasis. CLA expression is associated with the recruitment of CD4+ T cells into the skin, mainly when expressed on TCM cells, in particular on CD4+ TCM with the CCR4+ and CCR6+ phenotype [1,26]. In an earlier study, Bose et al. [32] proved that inhibition of the CCR7/CCL19 axis was critical for remission of psoriasis induced by TNF inhibitors [3]. In psoriasis, circulating CCR4 + CD4+ T cells significantly correlate with disease severity (PASI), and a strong negative correlation has been found for CCR5+ CD4+ T cells [32]. A subset of T effector CCR4+ CD8+ CD103+ cells also positively correlates with both systemic inflammation (CRP) and the severity of skin lesions [4,33]. It was also found that the fraction of IL-17 secreting CD4+ T-lymphocytes and probably γδ T-lymphocytes may play a role in the formation of a self-sustaining inflammatory loop [1].

Lymphocytes CD4+ Th producing pro-inflammatory cytokines, such as IL-17A, IL-22 and IFN-γ, are considered to be the main pathogenic T cell subpopulation, whereas CD8+, present in healthy skin as memory T cells, have a similar pro-inflammatory cytokine profile. They are in large quantities in the psoriatic epidermis and can recognize peptide antigens presented on MHC class I molecules like HLA-Cw6 [29]. Blockade of β1-integrin hinders entry of T cells into the epidermis, which prevents the development of disease eruptions on the murine AGR model of psoriasis [34]. Di Meglio et al. [29] showed that the accumulation of epidermal CD8+ cells induces both hyperproliferation of keratinocytes and papillomatosis (increase in CD8 + a parallel to the intensity of Ki67 staining in keratinocytes) [29]. CD8+ T cells isolated from psoriasis patients produce psoriasis-related cytokines. After treatment they remain in the skin as TRM and LL-37-specific CD8 + T cells, expressing integrin α1β1 [29].

In the study of Kurihara et al. [13], biopsies from psoriatic lesions were examined. The number of CD8+CD103+ TRM cells in the epidermis correlated with the thickness of the epidermis (*p* = 0.016), suggesting their role in the formation of psoriatic lesions [13]. They were mostly CD8+ CD69+ T cells, expressing skin colonization antigens, and the count of CD4+ CD103+ TRM was low. Some CD8+ CD103+ T cells produced IFN-γ, IL-17A or IL-22. In addition, CD8+ CD103+ TRM cells more frequently produced IL-17A than CD8+ CD103- and effector cells CD8+ CD103 or CD4+ CD103+. Interestingly, the number of CD8+ CD103+ IL-17A + TRM cells in patients treated with biological or systemic therapy was higher [13]. In contrast, in the dermis of the psoriatic plaques, TRM cells show lower expression of CD103, as demonstrated by Cheuk et al. [5]. In the first stage of psoriatic inflammation, in the skin without lesions yet, epidermal TRM cells expressing CCR6 cooperate with CCL20 expressing keratinocytes [35,36].

In turn, Vo et al. [36] evaluated the phenotypic features of TRM in non-lesional, lesional psoriasis and healthy skin. Immunofluorescence study showed that CD103+ CD8 TRM, both in non-lesional and lesional were dominant in the epidermis compared to the skin of healthy volunteers. In addition, IL-17A production was higher in patients with longer disease duration [36].

The optimal treatment time needed to completely silence TRM, which could ensure that there is no recurrence of lesions at the location is unknown [37]. Therefore, psoriatic lesions preferentially recur in previously affected areas of the skin, and pathogenic TRM cells exposed to IL-17A and IL-22 accumulate in resolved lesions [5,35]. In the study of Gallais Serezal et al. [38], tissue responses after T cell stimulation in healthy and psoriatic lesions were analyzed. An increase in the number of epidermal IL-17 and IL-22—producing skin-resident T CCR6 + cells—may be a genetically predisposed reaction to microbial stimulation in never-lesional skin in patients with psoriasis. The consequence is an increase in IFN-g production and stimulation of keratinocytes that release INF-a, which stimulates psoriatic inflammation.

At the same time psoriasis plaques showed IL-17-induced response patterns, indicating a relapse. The proportional amount of induced IFN-γ, IL-10e and IL-17A correlated with relapse time in patients after discontinuation of the treatment [38].

For the complete remission of the disease, full TRM suppression is required. Unfortunately, TRM cells are long-lived, and resistant to damaging factors and apoptosis. This explains the frequent relapses at the same location for psoriasis. Even after the clinical lesions have resolved, they are capable of producing IL-17A, and effective therapy only suppresses their activity [3,37]. Furthermore, CD8+ TRM cells accumulate in untreated psoriasis localizations, probably in correlation with disease duration [4,11]. An interesting issue is the explanation of the longevity of memory cells. One of the reasons is its resistance to apoptosis. The heterodimeric marker molecule CD8+ (αEβ7) consists of CD103 and β7 subunits and plays a key role in the stability of CD8+ TRM cells by increasing the level of the Bcl-2 molecule with anti-apoptotic activity [8,13]. In contrast, TRM can produce granzyme B (serine protease), which can induce cellular apoptosis [23]. Moreover, IL-15 probably plays an important role in TRM cell survival [24].

What is surprising, the effector T cells use glycolysis energy, which is less efficient in generating ATP, but faster. TCM, on the other hand, use endogenously synthesized fatty acids, glucose catabolism and oxidative phosphorylation to support their long-term survival and function [17,39]. TCMs use extracellular glucose from the blood to synthesize fatty acids in the endoplasmic reticulum, using lysosomal acid lipase, which is important in the hydrolysis of cholesteryl esters and triglycerides in LDL molecules to cholesterol and free fatty acids (FFA). Thanks to such mechanisms, they can be long-lived and able to react quickly to antigen [17]. Cui et al. additionally showed that IL-7, a cytokine critical for TCM differentiation and survival, induced glycerol transport and triacylglycerol synthesis via enhanced gene expression of glycerol channel aquaporin 9, thus providing substrates for mitochondria via fatty acid oxidation (FAO) [17,40].

Unfortunately, there is currently no detailed data on the metabolic properties of TRM. However, it is known that CD8+ TRM develop a transcriptional program with overexpression of FABP4 (adipocyte-FABP), FABP5 (epidermal-FABP), CD36 (a lipid scavenger receptor) and lipoprotein lipase (cleaves triglycerides to yield FFA and diacylglycerol). Such activity of genes responsible for the activity of these molecules is not found in naive T cells, TCM or effector memory cells [17].

The study by Pan et al. [17] showed that mouse CD8 + TRM cells generated by skin viral infection show different expression level of genes of proteins mediating the intracellular uptake and transport of lipids, including fatty acid-binding protein—FABP4 (adipocyte-FABP) and FABP5 (epidermal-FABP). These are molecules facilitating exogenous FFAs acquisition and metabolism [17]. CD8 + TRM upregulated the gene expression of FABP4/FABP5 in a peroxisome proliferator-activated receptor gamma (PPAR-γ)—dependent manner [17,41]. T-specific deficiency of these molecules has been shown to impair the uptake of exogenous FFA by CD8+ TRM cells, which significantly reduces their long-term survival in vivo, but does not affect the survival of TCM in lymph nodes. In addition, CD8+ TRM skin cells lacking FABP4/FABP5 were less effective in protecting mice against cutaneous viral infection [41]. It is interesting whether in patients with psoriasis the described situation will be similar to the one in healthy individuals. No studies have been published on this subject yet.

In vitro, CD8+ TRM cells showed increased oxidative metabolism of mitochondria in the presence of exogenous FFA in normal and psoriatic skin. These results suggest that FABP4 and FABP5 play a key role in the maintenance, longevity and function of CD8+ TRM cells, and suggest that CD8+ TRM cells use exogenous FFA and their oxidative metabolism to survive in tissue. It has been shown that the lack of FABP4 and FABP5 does not affect CD8+ T cell proliferation or skin recruitment, but they are necessary for their long-term survival in the skin [41]. Perhaps the blockade of PPAR-γ functions will allow the development of strategies for the treatment of psoriatic lesions.

Oxidative stress plays an important role in psoriatic inflammation. In the study of Esmaeili et al. [42], the redox status of TRM CD4+ and its correlation with IL-17 response were assessed. The increased intracellular ROS production in memory CD4+ T cells in psoriasis patients decreased catalase gene expression compared to healthy ones, but no differences in intracellular glutathione levels and plasma total antioxidant capacity were revealed. However, the above disorders did not affect the IL-17 response in memory T cells [42].

## 4. Tissue Resident Memory Cells in Psoriatic Arthritis (PsA)

Recently, data has shown that TRM cells are responsible for the development of synovitis in PsA. However, they were not observed in rheumatoid arthritis [23].

The latest study on psoriatic arthritis (PsA) and other spondyloarthritides (SpA) assessed the molecular profile, phenotype and function of synovial Tc17 cells (IL-17A + CD8 + T cells) to clarify their role in pathogenesis. It turned out that, as in psoriasis, they were mainly TCRαβ +, and their number was increased in synovial fluid vs. peripheral blood in patients with PsA or other SpA. In addition, synovial TA17 PsA cells had the characteristics of Th17 (RORC/IL23R/CCR6/CD161) and Tc1 cells (A/B granzyme). The synovial marker Tc17 was CXCR6, and elevated levels of CXCR6 CXCL16 ligand in PsA were found in synovial fluid, which could contribute to their retention in the joints [43]. However, there are many doubts if TRMs are responsible for the initiation of inflammation and whether it is the same phenotype that persists after the treatment [23].

A new theory has emerged that TRMs involved in synovitis migrate from the skin and gut. Although TRMs do not circulate in the blood, they can sometimes be activated so that they travel within the lymphatic system [23,44,45]. A study by Guggino et al. [45], assessing TRM cells in peripheral blood, gut and synovium in patients with SpA, showed that the expression of α4β7 may support the recirculation of these cells from the gut and peripheral blood to foci of inflammation in the joints. Although this study applies to the entire group of patients with ankylosing spondylitis, probably patients with PSA were also included. Unfortunately, PSA patients were not analyzed separately and there is no research on these patients.

## 5. The Role of Memory γδ T Cells

Matos et al. found that TRM in psoriasis patients had specific properties that TRM did not have in healthy skin. These pathogenic clones preferentially had Vβ and Vα in the TCR region. Researchers identified 15 TCRβ and 4 TCRα antigen receptor sequences in patients with psoriasis that were not seen in the skin of healthy patients or atopic dermatitis patients. In addition, they observed a reduced amount of γδ T cells in psoriasis compared to αβ T cells. It has been proven that IL-17- and IL-22-producing T-cell clones with IL-17- and IL-22-specific antigen receptors still exist in the skin after the active psoriatic lesions have resolved. Therefore, for full disease remission, suppression of these resident T lymphocyte populations is required [3].

However, in psoriasis a reduced amount of γδ T cells was observed compared to αβ T cells [3]. In contrast, γδ T lymphocytes, considered innate immune cells that can also give rise to TRM populations, are far less known. Recently it has been shown in a mouse model that they can produce IL-17A and IL-17F in the skin in response to imiquimod. Each subsequent test with imiquimod resulted in an ever stronger proliferation of γδ T cells compared to the skin of mice after the first provocation [46].

## 6. Th22 and Tc17 Lymphocytes

In active psoriasis, we observe increased expression of IL17A, IL22 and IFN-γ in the effector T CD4+ and CD8+ cells of the epidermis in the immediate vicinity of keratinocytes, whereas cutaneous T cells show less activity of these genes. Cheuk et al. [5] proved that Th22 epidermal cells in healed skin were still present and functioning after several years of disease remission. After their stimulation, an increase in local IL-22, IL-17A and IFN-γ production by CD4+ T cells was observed. The percentage of IL-22-producing CD4+ epidermal cells did not correlate with the number of years of treatment, which indicates a preserved effector function even after 6 years of the treatment [5]. It has been shown that CD4+ epidermal T cells produced mainly IL-22, which activates keratinocytes leading to acanthosis. In contrast, epidermal CD8+ T cells mainly produced IL-17A, which drives the production of pro-inflammatory cytokines and chemokines by keratinocytes and is involved in the recruitment of neutrophils [5].

Studies have also shown that there are T-cells producing both IL-22 and IL-17 in the epidermis, which are present in lesions and healed skin in patients receiving nb-UVB therapy, and are practically absent in patients treated with biological drugs. Perhaps this type of T cell is more pathogenic [5]. It is known that effective nb-UVB therapy induces T cell apoptosis in the skin, but highly active effector lymphocytes can be constantly recruited from circulation [5].

Of note, active effector T cells involved in the inflammatory process may develop into long-lived epidermal TRM cells as the disease subsides [5,7,8].

## 7. Epidermal Langerhans Cells (LCs)

Psoriatic lesions’ “memory” may also be affected by Langerhans cells dysfunction. In normal skin, epidermal Langerhans cells (eLC) are located in the epidermis and separated from dendritic skin cells (DC) by the basal membrane. Compared with cutaneous DC, LCs express fewer Toll-like receptors (TLRs), indicating an impaired ability to respond to TLR signaling, probably to maintain a state of tolerance to commensal [47]. In active psoriasis, under the influence of IL-23 and IL-17, inflammatory DCs penetrate both to the epidermis and dermis. In inflammatory DC epidermal expression of CCR2+ (eDC) occurs only in psoriasis and is phenotypically different from normal DCs’. Their number is higher than eLC and they show high expression of genes involved in the recruitment of neutrophils, keratinocytes and T lymphocytes. On the other hand, colonization of epidermis by eLC, under the influence of TLR-4 and TLR-7/8 activation, produce large amounts of IL-23, and eDCs secrete IL-1β together with IL-23 and TNF-α. After the psoriasis eruption, eDCs are absent in unchanged skin, while eLC retain high IL-23A expression and continue to respond to TLR stimulation, producing IL-23 [47,48]. Although eDCs are 3–10 times more numerous than eLC and drive the psoriatic condition, they also have the ability to produce anti-inflammatory IL-10 [48].

In individuals with early type of psoriasis (under 40 years of age) LCs do not show the ability to migrate under the influence of stimuli (chemical allergens, IL-1b and TNF-α). It is probably a phenomenon caused by the influence of the psoriatic environment rather than their defect [49,50]. Their ability to migrate is restored under the influence of treatment (among others anti-TNF, anti-IL-12/23, fumaric acid esters and UVB), except for therapies directed mainly towards T cells (ciclosporin A, MTX) [50,51,52]. Therefore, the observed rapid relapses after discontinuation of ciclosporin A may be explained by the role of LC in the rapid initiation of the inflammation in the same location (effect on TRM expressing IL-23R), which thus affects the memory of psoriatic eruptions [48].

## 8. Clinical Significance of the Memory of Psoriasis Lesions

Psoriatic lesions are characterized by the presence of massive inflammatory infiltrates (lymphocytes, neutrophils), proliferation and disturbance of epidermal differentiation (parakeratosis) as well as the formation of new blood vessels. Despite being such large structures, their resolution leaves no scars. This is quite unusual, especially comparing to other conditions in which neutrophils (pyoderma gangrenosum) are involved, where scars are present [12]. However, after discontinuation of treatment, a rapid recurrence of lesions is usually observed (90% in the same location) [24]. The understanding of the psoriasis context and the role of TRM can explain the key issues associated with this disease:-the role in resistance to treatment and reactivation of changes in the same location [24];-the role in the Koebner isomorphic symptom [24];-the decision about adequate treatment time to limit lesions relapse and withdraw and reduce the amount of TRM.

The clinical characteristics of recurrence of psoriatic lesions mirrors the biology of TRMs. Since TRM does not work, inflammatory lesions induced by TRM are usually well demarcated with clear margins. The relapse of lesions can be fast because they occur in TRM skin and cooperate with local dendritic cells [12,52]. In understanding the abovementioned mechanisms, one stands a chance to provide more effective treatment and thus long-lasting remissions in psoriasis patients.

## 9. Conclusions

The discovery of TRM cells in psoriasis allows a better understanding of the complicated relationships between mediators and cells of the innate and adaptive immune system, keratinocytes and endothelial cells. It explains the clinical phenomenon of the tendency of psoriatic lesions to relapse in the same location and it allows to develop new therapeutic strategies in the future.

## Figures and Tables

**Figure 1 ijms-21-00625-f001:**
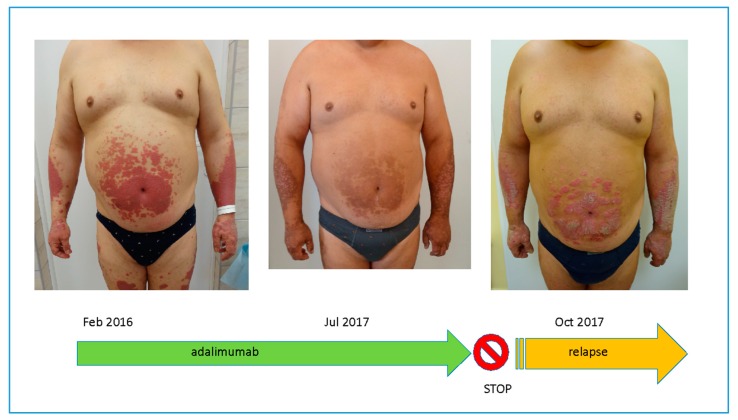
The relapse of psoriatic lesions in the same localization despite efficient treatment.

**Figure 2 ijms-21-00625-f002:**
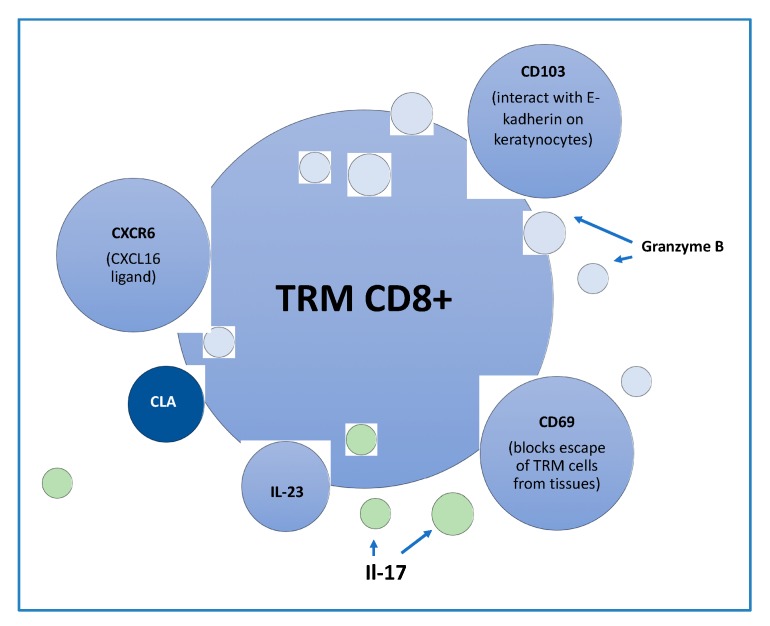
The receptors and secretory activity of tissue resident memory (TRM) CD8+ [5,22]

**Figure 3 ijms-21-00625-f003:**
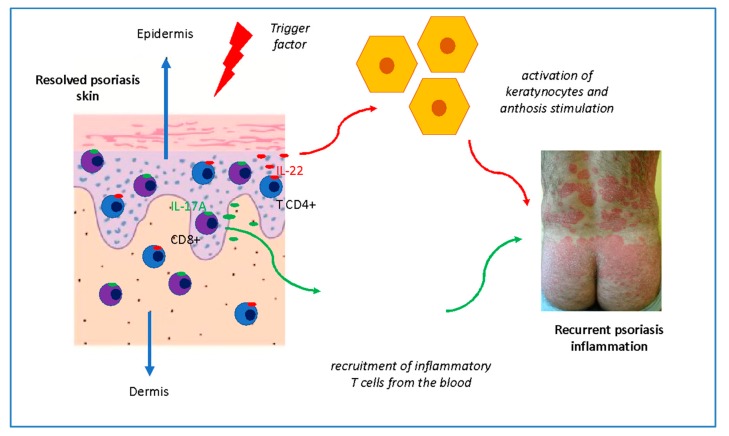
The role of TRM in psoriatic inflammation.

**Table 1 ijms-21-00625-t001:** The Cells Responsible for the “memory” of Psoriatic Lesions.

Type of Cells		Psoriasis Lesions	Resolved Psoriasis Skin
**TRM**	**CD4+**	A few in dermis CD103- and a low number in epidermis CD103+	A few
	**CD8+**	A lot of in epidermis CD103+: CD8+CD49a+ producing perforin, IFN-γ CD8+CD49a- producing IL-17 IL-22, IFN-γ, a few in dermis CD103- and a low number CD103+	A few in epidermis
**Dendritic Cells**	**LC**	In epidermis producing a lot of IL-23	In epidermis producing IL-23
	**eDC**	In epidermis producing of IL-1β, IL-23, TNFα	Absent

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
