# Peer review of "Immunological Memory of Psoriatic Lesions"

_ijms, 2020, doi:10.3390/ijms21020625_

Round 1

Reviewer 1 Report

Congratulations to the authors for a very thorough intensive review explaining the complex mechanisms ofclinical phenomenon of tendency of psoriatic lesions to relapse in the same locations via tissue resident memory cells (TRM). The manuscrpit points to promising paths for the development of newer therapeutical strategies in the future. Excellent paper suitable for the publication in it current form.

Author Response

Dear Reviewer,

Thank you very much for many valuable comments and a lot of time spent on our manuscript. Thanks to this, I could complete my work.

Best regards,

Agnieszka Owczarczyk-Saczonek

Reviewer 2 Report

Comments for author:

The topic of the review by OwczrczyK-Saczonek et al is very actual and interesting. The review is well written at the beginning, but there are some imprecisions during the writing, and some unclear parts. Moreover, some of the reported results from literature are sometimes uncorrect.

Major points are:
1) Page 2 lane 71, recirculation of TRM in unusual situations (what are these situations?). A reference will be better.
2) Paragraph Tissue Resident cells in psoriasis: lane 101: the paper by Lande et al Nat comm. 2014 states that also CD4 T-cells recognize LL37 as an autoantigen and their correlate with PASI. This should be mentioned.
3) Lane 1015 -107: this statement seems to be a repetition of lane 67-69.
4) Lane 110, reference is lacking regarding the role of IL-15 and IL-7 in recruitment.
5) Lane 117, TRMs have been described in unchanged (means uninvolved or never involved skin?).
6) Lane 148 CD8 in healthy skin, why is this mentioned if the author are talking about psoriasis T-cells?
7) Lane 155-157: the statement is not clear, the paper ref 33 cannot talk about LL37 specific cells, discovered in 2014.
8) Lane 158, Kurihara et al (ref to be added, 13).
9) Lane 159, its number should be substituted with their number correlate…..
10) Lane 169, ref is lacking, still 13?
11) Lane 177-179, this statement is not clear. Should be written again.
12) Lane 184, lymphomas are introduced but there are not pertinent (a better explanation is needed)
13) Lane 208: Pan et al : Reference needed, 17
14) Lale 210-217 are refereed to Healthy individuals. This is a bit confusing….what about psoriasis patients?
15) Lane 225: reference is lacking, no. 40
16) The study mentioned at lane 245 is on ankylosing spondylitis only, this should be clarified and may possibly apply to PSA, but no PSA patients were analysed.
17) Lane 266: lacl of reference (no. 5?)
18) The following paper could be discussed:
A skewed pool of resident T cells triggers psoriasis-associated tissue responses in never-lesional skin from patients with psoriasis, Volume 143, Issue 4, April 2019, Pages 1444-1454. Journal of Allergy and Clinical Immunology

Author Response

The response to reviewer

Dear Reviewer,

Thank you very much for many valuable comments and a lot of time spent on our manuscript. Thanks to this, I could complete my work.

Best regards,

Agnieszka Owczarczyk-Saczonek

Comments for author:

The topic of the review by Owczarczyk-Saczonek et al is very actual and interesting. The review is well written at the beginning, but there are some imprecisions during the writing, and some unclear parts. Moreover, some of the reported results from literature are sometimes uncorrect.

Major points are:
1) Page 2 lane 71, recirculation of TRM in unusual situations (what are these situations?). A reference will be better.- S1PR1 reacts to the S1P ligand, which is released by endothelial cells in the blood and lymph, allowing TRM cells to leave the tissues for circulation. This situation is observed, among others after exposure to an allergen resulting in formation of secondary TRM cells. [Behr, F. M., Chuwonpad, A., Stark, R., & van Gisbergen, K. (2018). Armed and Ready: Transcriptional Regulation of Tissue-Resident Memory CD8 T Cells. Frontiers in immunology9, 1770. doi:10.3389/fimmu.2018.01770]
2) Paragraph Tissue Resident cells in psoriasis: lane 101: the paper by Lande et al Nat comm. 2014 states that also CD4 T-cells recognize LL37 as an autoantigen and their correlate with PASI. This should be mentioned.

However, in recent years, the role of CD8+ T cells, autoreactive against cathelicidin-specific protein LL37 (antimicrobial peptide derived from keratinocytes) and ADAMTSL5 (protein produced by melanocytes) in the pathogenesis of the disease has been more appreciated. The abovementioned proteins are considered as autoantigens in psoriasis. Similarly, CD4 T-cells recognize LL37 as an autoantigen and they correlate with PASI. [Lande, R., Botti, E., Jandus, C. et al. The antimicrobial peptide LL37 is a T-cell autoantigen in psoriasis. Nat Commun 5, 5621 (2014) doi:10.1038/ncomms6621].
3) Lane 1015 -107: this statement seems to be a repetition of lane 67-69. – I’d like to emphasize the role of CD69 in the context.
4) Lane 110, reference is lacking regarding the role of IL-15 and IL-7 in recruitment.- Adachi, T., Kobayashi, T., Sugihara, E., Yamada, T., Ikuta, K., Pittaluga, S., … Nagao, K. (2015). Hair follicle-derived IL-7 and IL-15 mediate skin-resident memory T cell homeostasis and lymphoma. Nature medicine21(11), 1272–1279. doi:10.1038/nm.3962

5) Lane 117, TRMs have been described in unchanged - uninvolved not never involved skin (after in healed skin).
6) Lane 148 CD8 in healthy skin, why is this mentioned if the author are talking about psoriasis T-cells? - to highlight the potential of TRM to initiate inflammation (lost citation of Di Meglio)
7) Lane 155-157: the statement is not clear, the paper ref 33 cannot talk about LL37 specific cells, discovered in 2014.- lost citation of Di Meglio
8) Lane 158, Kurihara et al (ref to be added, 13).-I’ve done it.
9) Lane 159, its number should be substituted with their number correlate…..The number of CD8+CD103+ TRM cells in the epidermis correlated with the thickness of the epidermis (P = 0.016) suggesting their role for the formation of psoriatic lesion [Kurihara]
10) Lane 169, ref is lacking, still 13? [39], I’ve done it.
11) Lane 177-179, this statement is not clear. Should be written again.

An increase in the number of epidermal Il-17 and IL-22 – producing skin-resident T CCR6 + cells may be a genetically predisposed reaction to microbial stimulation in never-lesional skin in patients with psoriasis. The consequence is an increase in IFN-g production and stimulation of keratinocytes that release INF-a, which stimulates psoriatic inflammation.

Gallais Sérézal I, Hoffer E, Ignatov B, Martini E, Zitti B, Ehrström M, Eidsmo L. A skewed pool of resident T cells triggers psoriasis-associated tissue responses in never-lesional skin from patients with psoriasis. Journal of Allergy and Clinical Immunology, Volume 143, Issue 4, 1444 - 1454

12) Lane 184, lymphomas are introduced but there are not pertinent (a better explanation is needed) - Indeed lymphomas are not relevant here, I crossed out.
13) Lane 208: Pan et al : Reference needed, 17 - I’ve done it.
14) Lane 210-217 are refereed to Healthy individuals. This is a bit confusing….what about psoriasis patients? It is interesting whether in patients with psoriasis the described situation will be similar to the one in healthy individuals. No studies have been published on this subject yet.
15) Lane 225: reference is lacking, no. 40- I’ve done it.
16) The study mentioned at lane 245 is on ankylosing spondylitis only, this should be clarified and may possibly apply to PSA, but no PSA patients were analysed.- I added an explanation

Although this study applies to the entire group of patients with ankylosing spondylitis, probably patients with PSA were also included. Unfortunately, PSA patients were not analyzed separately and there is no research on these patients.
17) Lane 266: lack of reference (no. 5?) - I’ve done it.
18) The following paper could be discussed:
A skewed pool of resident T cells triggers psoriasis-associated tissue responses in never-lesional skin from patients with psoriasis, Volume 143, Issue 4, April 2019, Pages 1444-1454. Journal of Allergy and Clinical Immunology  - I’ve added it.
